

# TLR4 promoter rs1927914 variant contributes to the susceptibility of esophageal squamous cell carcinoma in the Chinese population

Jiaying Li[1,2], Hongjiao Wu[1], Hui Gao[1], Ruihuan Kou[3], Yuning Xie[1,2], Zhi Zhang[3] and Xuemei Zhang[1,2]

[1] School of Public Health, North China University of Science and Technology, Tangshan, China
[2] College of Life Science, North China University of Science and Technology, Tangshan, China
[3] Affliated Tangshan Gongren Hospital, North China University of Science and Technology, Tangshan, China

Corresponding author
Xuemei Zhang, jyxuemei@gmail.com

## ABSTRACT

**Background**. Toll-like receptor 4 (TLR4), as a key regulator of both innate and acquired immunity, has been linked with the development of various cancers, including esophageal cancer. This study aims to analyze the association of potential functional genetic polymorphisms in TLR4 with the risk of esophageal cancer.

**Methods**. This case-control study involved in 480 ESCC patients and 480 health controls. Polymerase chain reaction-restriction fragment length polymorphism (PCR-RFLP) was used to genotype TLR4 rs1927914 polymorphism. Taqman probe method was used to determine the genotypes of TLR4 rs11536891 and rs7873784 variants. The relationship between TLR4 genetic variation and ESCC risk was analyzed by Logistic regression model by calculating the odds ratio (OR) and 95% confidence interval (95% CI).

**Results**. Compared with TLR4 rs1927914 AA genotype carriers, GG carriers had a lower ESCC risk (OR = 0.59, 95% CI [0.38–0.93], P = 0.023). Stratification analysis by age showed that TLR4 rs1927914 GG could affect the risk of ESCC in elderly people (OR = 0.59, 95% CI [0.36–0.97]). Smoking stratification analysis indicated that rs1927914 GG carriers were related to ESCC susceptibility among non-smokers (OR = 0.36, 95% CI [0.18–0.73]). Dual luciferase reporter assay suggested that rs1927914 G-containing TLR4 promoter displayed a 1.76-fold higher luciferase activity than rs1927914 A-containing counterpart in KYSE30 cells. Electrophoretic mobility shift assay (EMSA) showed the KYSE30 cell nuclear extract was able to bind the probe with rs1927914 G allele and this DNA-protein interaction could be eliminated by competition assays with unlabeled rs1927914 G probe, which indicating that the binding is sequence-specific. Our results also showed that TLR4 rs7873784 (G>C) and rs11536891 (T>C) conformed to complete genetic linkage. The genotype distributions of TLR4 rs11536891 variant among ESCC patients and normal controls have no statistical significance.

**Conclusion**. The TLR4 rs1927914 variant contributes to the ESCC risk by effecting the promoter activity.

## INTRODUCTION

Esophageal cancer, as the sixth leading cause of cancer death, is one of the most common malignant tumors worldwide (*Bray et al., 2018*). Esophageal cancer contains two common histological types: esophageal adenocarcinoma (EAC) and esophageal squamous cell carcinoma (ESCC). There are clear differences between EAC and ESCC that affect their distribution and incidence in the world (*Domper Arnal, Ferrández Arenas & Lanas Arbeloa, 2015*; *Yang, Chen & Tu, 2016*). In China, most of the cases of esophageal cancer are squamous cell cancer (*Lin et al., 2013*). ESCC is caused by environmental and genetic factors. Epidemiological studies have reported that tobacco smoking, alcohol drinking, ingesting hot substances and so on played a role in the development of ESCC (*Yu et al., 2018a*). However, not all individuals who have been exposed to these hazards eventually get ESCC. In recent years, genetic polymorphisms have been reported to impact the development of esophageal cancer (*Hiyama et al., 2007*; *Yue et al., 2017*).

Single nucleotide polymorphism (SNP) is one of the most common genetic variants in the genome. Over the past decade, large-scale SNP analyses, known as genome-wide association studies (GWAS), have provided a new way to identify genetic loci which might be associated with the cancer susceptibility, survival prognosis or drug response (*Wu et al., 2013*; *Yu et al., 2018b*; *Zhang et al., 2020*). The SNPs located in specific genes, which involved in cancer-related pathway, may modulate gene expression or protein activity and further involved in cancer initiation and development. For example, the functional genetic variants in cyclooxygenase-2 and 12-lipoxygenase have been reported to be associated with the risk of esophageal cancer (*Guo et al., 2007*; *Zhang et al., 2005*). The mutations in Flap endonuclease 1 (Fen1), which is one of key components in long-patch DNA base-excision repair, resulted in autoimmunity, chronic inflammation and various cancers (*Zheng et al., 2007*).

The interaction between the immune system and malignant cells has an impact on tumorigenicity (*Terme & Tanchot, 2017*). On one hand, the immune system kills or clears malignant transformed cells; on the other hand, malignant cells struggle to escape immune surveillance (*De Visser, Eichten & Coussens, 2006*; *Schreiber, Old & Smyth, 2011*). As the most studied pattern recognition receptor, Toll-like receptors (TLRs) can enhance the innate immune response and stimulate antigen-derived cells such as dendritic cells, and then activate the tumor-specific T cells immune which involving in the development of tumors (*Kaczanowska, Joseph & Davila, 2013*; *Pham et al., 2010*). TLR4 can not only recognize extracellular antigens, but also respond to intracellular injury related factors (*Jacobsen, Aasenden & Hensten-Pettersen, 1991*; *Rocha et al., 2016*). A study showed that TLR4 induced by LPS promoted the secretion of immunosuppressive cytokines which promoted the proliferation of lung cancer and ESCC cells (*He et al., 2007*; *Zu et al., 2017*). TLR4 also involved in the antitumor T-cell immune response by induced by danger-associated molecular patterns (DAMPs) (*Fang et al., 2014*). Studies have shown that TLR4 is overexpressed in a variety of malignant tumors and associated with poor prognosis in cancer patients (*Li et al., 2017*; *Pandey, Chauhan & Jain, 2018*; *Sheyhidin et al., 2011*; *Wang*

*et al., 2017*; *Zhao et al., 2019*). TLR4 has been identified as a potential drug target for the immuno-therapeutics in various cancers (*Shetab Boushehri & Lamprecht, 2018*).

In view of the important role of TLR4 in tumors, we screened out the potential functional SNPs in TLR4 using bioinformatic methods and then performed a case-control study in Chinese population to determine whether they were correlated with the occurrence of ESCC.

## MATERIALS AND METHODS

### Study subjects

In this study, 480 ESCC patients and 480 cancer-free controls were included. Cases were recruited from Apr 2008 to Dec 2012 in Affiliated Tangshan Gongren Hospital and Tangshan Renmin Hospital of North China University of Science and Technology (Tangshan, China). Inclusion criteria: all patients were diagnosed as primary ESCC by histopathology; all specimens were genetically unrelated Han Chinese; none of the patients had received radiotherapy or chemotherapy. Four hundred and eighty healthy individuals were randomly recruited from the same region and matched with cases on age and sex. All participants signed the written informed consent. Institutional Review Board of North China University of Science and Technology had approved the research (12-002).

### TLR4 SNPs selection

In this study, we predicted the possible functional SNPs in the regulatory region of TLR4. All included SNPs located in the promoter region or the 3′ untranslated region with MAF ≥0.05. For SNPs in the promoter region of TLR4, transcription factor binding capability was predicted by TRANSFAC program (*Wingender et al., 1996*). For the SNPs located in the 3′ untranslated region, microRNA binding ability was predicted using SNPinfo Web Server (*Xu & Taylor, 2009*). Finally, TLR4 rs1927914 in the promoter region and rs11536891 and rs7873784 in the 3′ untranslated region were selected for further genotyping (Fig. 1A).

### Genotype of selected TLR4 polymorphisms

Each subject donated 2 mL of peripheral blood. DNA was extracted using the blood DNA kit provided by TIANGEN Biotech (Beijing). TLR4 rs1927914 genotyping was performed by the Polymerase chain reaction-restriction fragment length polymorphism (PCR-RFLP). The target DNA fragment was amplified by PCR using the forward primer 5′-TGACATGGAAAATGGAGAGATAGAGG-3′and reverse primer 5′-GGACTATGATGGAGATTGAAAATGTGG-3′. PCR was performed using a 6μl reaction system containing 0.05 μM each primer, 10ng DNA, and 2 x Es Taq MasterMix (CWBIO, Beijing, China). PCR procedure was 3 min at 95 °C, followed by 32 cycles (30s at 95 °C, 30s at 56.5 °C and 34s at 72 °C) and 5 min at 72 °C for final extension. TLR4 PCR products were cut by Nsi I and verified with 3% agarose gel. TLR4 rs11536891 and rs7873784 variants were genotyped by SNP genotyping assays (C_31784036_10 and C_29292008_10) (Thermo Fisher Scientific, Waltham, USA). TaqMan SNP assay includes two allele-specific TaqMan MGB probes and a PCR primer pair that uniquely amplify the region flanking of SNP. The MGB probes do not fluoresce because of the non-fluorescent quencher (NFQ) at

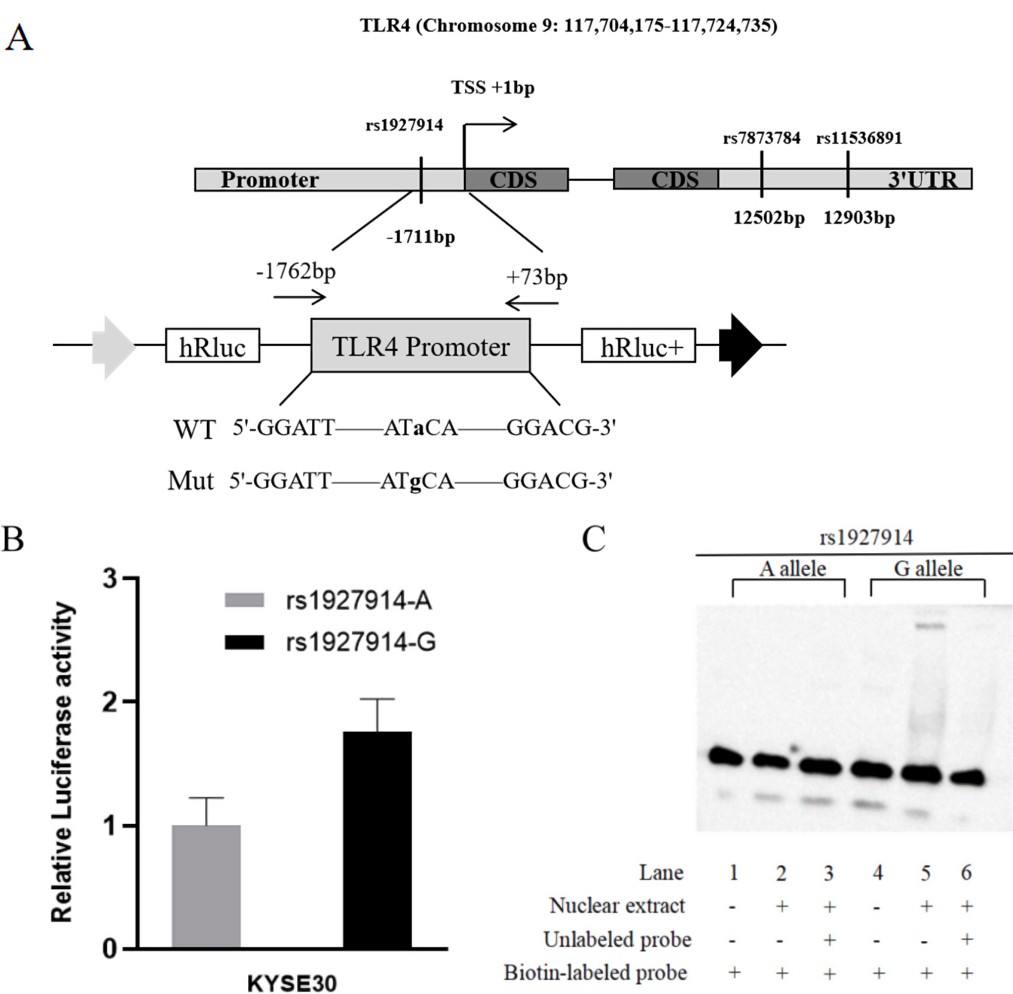

**Figure 1   TLR4 locus with SNPs and the functional analysis of rs1927914.** (A) A schematic showing TLR4 locus with candidate SNPs. (B). Luciferase expression of two constructers (pGL3- rs1927914G and pGL3- rs1927914A) in KYSE30 cells co-transfected with pRL-SV40 to standardize the transfection efficiency. Luciferase levels of pGL3-Basic and pRL-SV40 were determined in triplicate. Fold increase was measured by defining the activity of the empty pGL-3 Basic vector as 1. *$P < 0.05$. (C). Electrophoretic mobility shift assays with biotin-labeled oligonucleotide probes containing TLR4 rs1927914A or G allele. Lanes 1 and 4 show the gel mobilities of the labeled probes without nuclear extracts; lanes 2 and 5 show the mobilities of the labeled probes with nuclear extracts in the absence of competitor; and lanes 3 and 6 show the mobilities of the labeled probes with nuclear extracts and unlabeled competitors. The arrow localizes the major probe-nuclear protein complex.

the 3′ end of the Taqman probe. Two allele-specific probes contain different reporter dyes (FAM and VIC) specifically hybridize to the allele specific sequence. The 5′ nuclease activity of AmpliTaq Gold DNA polymerase in TaqMan Genotyping Master Mix (Thermo Fisher Scientific, Waltham, MA, USA) can only cleave the hybridized probes. This will separate the reporter dye from the quencher and allow fluorescence emission and be detected.
## Vector construction and site-directed mutation

To analyze the effect of TLR4 promoter region genetic variation on transcriptional activity, we constructed a reporter plasmid containing -1762 to +73 base pairs of human TLR4 promoter. The primers used to amplify this fragment were 5′-GGGGTACCCCGGATTGGAAGTGCTTGGAG-3′ and 5′-CTAGCTAGCTAGAAGAAGAAAACGCCTGC-3′, which contain Kpn I and Nhe I recognition site (underlined sequence) in forward primer and reverse primer, respectively (Fig. 1A). The PCR product was then cloned into pGL3-basic reporter vector (Promega, Madison, WI, USA). Based on the sequence results, we constructed pGL3-rs1927914A-containing plasmid. The template vectors (pGL3- rs1927914 A) were then used to obtain pGL3-rs1927914G-containing vector by site-specific mutagenesis reaction using site-specific mutation kit (TIANGEN, Beijing, China). All constructs were verified by direct sequencing.

## Cell culture, Transfection and luciferase assay

Esophageal carcinoma cells (KYSE30) were kindly gifted from Dr. Y. Shimada in Hyogo College of Medicine (Japan). Cells were cultured in DMEM medium containing 10% FBS (Gibco, Vienna, Austria) and 1% penicillin and streptomycin. Cells were seeded at a density of $3 \times 10^5$ cells/well in 24-well plate to 70–80% confluence. Cells were co-transfected with different pGL3-Basic vectors and pRL-SV40 using Lipofectamine[TM] 2000 (Invitrogen, Carlsbad, USA). Luciferase activity was detected by Dual Luciferase Reporter Assay. A 13 μL of cell lysate was mixed with 25 μL of Luciferase Assay Reagent II, and Firefly luciferase activity was measured by GloMax 20/20 Luminometer. Then, 25 μL of 1×Stop & Glo solution was added to determine Renilla luciferase activity. The ratio of Firefly and Renilla luciferase activity was presented to the level of relative luciferase activity. Independent experiments were performed three times.

## Electrophoretic mobility shift assay (EMSA)

The biotin-labeled oligonucleotide probes (5′-TCTAGGACTTAGCATACAAATATTCCTGTT-3′and 5′-TCTAGGACTTAGCATGCAAATATTCCTGTT-3′) containing TLR4 rs1927914 A/G allele was synthesized by Sangon Biotech (Shanghai, China). Nuclear proteins were extracted from KYSE30 cells by using NE-PER[TM] Nuclear and Cytoplasmic Extraction Reagents (Thermo Fisher Scientific, Waltham, MA, USA). The electrophoretic mobility shift assays were conducted by using the LightShift[TM] Chemiluminescent EMSA kit (Thermo Fisher Scientific, Waltham, MA, USA) following the instruction from manufacturer. Briefly, each 20fmol labeled oligonucleotide was incubated with 8μg nuclear extract for 10 min in 1× binding solutions. For competition experiment, we added 4pmol unlabeled oligonucleotide probe before incubating with labeled probe. After electrophoresis in a 6.5% polyacrylamide gel, the electrophoresed binding reactions were transferred to positively charged nylon membrane and then were crosslinked by UVJLY-1 UV-light crosslinking instrument of JIAYUAN Industrial Technology (Beijing, China). Biotin-labeled DNA was then detected and visualized by Luminol/Enhancer Solution and Stable Peroxidase Solution in LightShift[TM] Chemiluminescent EMSA kit.

## Statistical analysis

In this study, all the research data were statistically analyzed using SPSS 23.0 (SPSS, Chicago, USA). The differences of basic characteristics in cases and controls were tested by $\chi^2$ test. The Hardy–Weinberger equilibrium (HWE) of TLR4 polymorphisms in controls were tested by $\chi^2$ test. The correlation between the genetic variants in TLR4 and the risk of esophageal cancer were evaluated by *OR* and 95% CI. The activity of luciferase reporter gene was compared by two independent sample *t*-test. $P < 0.05$ indicated statistically significant. Linkage disequilibrium (LD) analysis was performed by HaploReg (*Ward & Kellis, 2012*).

# RESULTS

## Study subjects' general demographic characteristics

The general information of all subjects was showed in Table 1. There were no significant differences in age and gender between the cases and controls ($P > 0.05$). The proportion of smokers in the case group was 64.4% and in control group was 30.6% ($P < 0.001$), indicating a statistical difference. However, there were no statistically significant differences in cumulative smoking among ESCC patients and healthy controls ($P = 0.149$).

## The influence of TLR4 variants on ESCC risk

After predicted by TRANSFAC program and SNPinfo Web Server, three potential functional SNPs (rs1927914, rs7873784, rs1536891) were selected for further analysis (Table 2). After genotyping TLR4 rs7873784 polymorphism in 100 samples, we found that the frequencies of GG, GC and CC genotype were 87.0%, 12.0% and 10% which is the same as that of TT, CT and CC genotype of rs11536891 variant. We then measured the amount of linkage disequilibrium (LD) and demonstrated that two TLR4 SNPs (rs7873784 and rs11536891) conformed to complete genetic linkage with D' of 1.00 and $r^2$ of 1.00. Based on this, in further study, we only genotyped TLR4 rs11536891 and rs1927914 polymorphisms. Table 3 showed the association of TLR4 rs1927914 and rs11536891 genotypes with the susceptibility to esophageal cancer. Genotypes distribution of 2 SNPs among controls group were consistent with the Hardy–Weinberg equilibrium (HWE), indicating that the selected population was well representative. The genotypes frequencies of TLR4 rs1927914 AA, GA and GG were 40.6% (195), 49.4% (237) and 10% (48) in cases and 35.2% (169), 49.6% (238) and 15.2% (73) in controls. Multivariate logistic regression analysis displayed that rs1927914 GG genotype contributed to a decrease ESCC risk ($OR = 0.59$, 95% CI [0.38–0.93], $P = 0.023$) when compared with AA genotype. There was no significant difference in the distribution of TLR4 rs11536891 genotypes in the case group and the control group ($P > 0.05$).

## Stratification analysis

The stratification analysis by gender, age and smoking status was used to further explore the interaction effect of genetic variation of TLR4 rs1927914 on ESCC risk (Table 4). When stratified by gender, there was no significant correlation between genotypes of TLR4 rs1927914 and the esophageal cancer risk among males and females ($OR = 0.67$, 95%

**Table 1   Distributions of select characteristics in cases and control subjects.**

| Variables | Case (n = 480) | | Controls (n = 480) | | P value[a] |
|---|---|---|---|---|---|
| | No | (%) | No | (%) | |
| Sex | | | | | 0.930 |
| Male | 403 | 84.0 | 402 | 83.7 | |
| Female | 77 | 16.0 | 78 | 16.3 | |
| Age | | | | | 0.162 |
| ≤50 | 83 | 17.3 | 100 | 20.8 | |
| >50 | 397 | 82.7 | 380 | 79.2 | |
| Smoking status | | | | | <0.001 |
| Non-smoker | 171 | 35.6 | 333 | 69.4 | |
| Smoker | 309 | 64.4 | 147 | 30.6 | |
| Pack year of smoking | | | | | |
| ≤25 | 123 | 39.8 | 69 | 46.9 | 0.149 |
| >25 | 186 | 60.2 | 78 | 53.1 | |

**Notes.**
[a]Two-sidde $\chi^2$ test.

**Table 2   General information of 3 SNPs of TLR4.**

| SNP | Location | Allele | MAF | Functional changes |
|---|---|---|---|---|
| rs1927914 | promoter region | A/G | 0.49 | Oct-1 |
| rs7873784 | 3′UTR | G/C | 0.14 | hsa-miR-144 |
| rs11536891 | 3′UTR | T/C | 0.14 | hsa-miR-519a, hsa-miR-519b-3p |

**Table 3   Gene polymorphism of TLR4 and their association with ESCC.**

| TLR4 genotypes | Cases (n = 480) | | Controls (n = 480) | | OR (95% CI) | P value[a] |
|---|---|---|---|---|---|---|
| | No | (%) | No | (%) | | |
| Rs1927914 | | | | | | |
| AA | 195 | 40.6 | 169 | 35.2 | | |
| GA | 237 | 49.4 | 238 | 49.6 | 0.91(0.68–1.22) | 0.528 |
| GG | 48 | 10.0 | 73 | 15.2 | 0.59(0.38–0.93) | 0.023 |
| Rs11536891 | | | | | | |
| TT | 410 | 85.4 | 410 | 85.4 | | |
| CT | 64 | 13.3 | 68 | 14.2 | 0.96(0.65–1.43) | 0.847 |
| CC | 6 | 1.3 | 2 | 0.4 | 4.59(0.87–24.25) | 0.073 |

**Notes.**
[a]Data were analyzed by unconditional logistic regression and adjusted for sex, age and smoking status.

CI [0.41–1.09]; $OR = 0.31$, 95% CI [0.09–1.11]). In the age stratification, median age (50-year) in controls was set as cut-off value for all subjects. Our data showed that older subjects (age > 50) with GG genotype had a lower esophageal cancer risk than those with the AA genotype ($OR = 0.59$, 95% CI [0.36–0.97]), but the younger subjects didn't ($OR = 0.53$, 95% CI [0.18–1.55]). In a stratified analysis based on smoking status, we found

**Table 4  Stratified analysis between TLR4 rs1927914 genotypes and ESCC risk.**

| Variables | Genotypes (Cases/Controls) | | | GG/AA model | GA/AA model |
|---|---|---|---|---|---|
| | AA | GA | GG | OR (95% CI)[a] | OR (95% CI)[a] |
| Sex | | | | | |
| Male | 195/142 | 237/197 | 48/63 | 0.67(0.41–1.09) | 0.95(0.69–1.32) |
| Female | 35/27 | 38/41 | 4/10 | 0.31(0.09–1.11) | 0.73(0.37–1.44) |
| Age | | | | | |
| ≤50 | 38/33 | 34/52 | 11/15 | 0.53(0.18–1.55) | 0.55(0.26–1.17) |
| >50 | 157/136 | 203/186 | 37/58 | 0.59(0.36–0.97)* | 1.00(0.73–1.38) |
| Smoking status | | | | | |
| Non-smoker | 73/109 | 86/170 | 12/54 | 0.36(0.18–0.73)* | 0.76(0.51–1.13) |
| Smoker | 122/60 | 151/68 | 36/19 | 0.93(0.49–1.76) | 1.12(0.73–1.71) |
| Pack year of smoking | | | | | |
| ≤25 | 49/28 | 61/35 | 13/6 | 1.26(0.43–3.68) | 0.98(0.53–1.84) |
| >25 | 73/32 | 90/33 | 23/13 | 0.78(0.35–1.74) | 1.26(0.70–2.26) |

**Notes.**
[a]Data were analyzed by unconditional logistic regression and adjusted for sex, age and smoking status.
*$P < 0.05$.

that the GG genotype was a protective factor among non-smoker ($OR = 0.36$, 95% CI [0.18–0.73]), but not among smoker ($OR = 0.93$, 95% CI [0.49–1.76]).

## Luciferase reporter gene activity detection

For further verification, we assessed the effect of TLR4 rs1927914 genetic variation on transcriptional activity. We transiently transfected the recombinant plasmid with rs1927914A (pGL3-Basic-A), G allele (pGL3-Basic-G) or pGL3-Basic into KYSE30 cells together with an internal control plasmid to detect the expression of luciferase activity, respectively. The results showed that luciferase activity drived by TLR4 rs1927914 G allele was 1.76-fold higher than that by rs1927914 A allele ($P = 0.0043$) (Fig. 1B).

## Allele-specific binding of nuclear proteins to TLR4 promoter

We conducted the electrophoretic mobility shift assay to investigate if different TLR4 rs1927914 allele effected on the binding activity to transcriptional factor. Biotin-labeled probes containing two different alleles (rs1927914 A and G) were respectively reacted with the KYSE30 nuclear extract. As showed in Fig. 1C, rs1927914G-protein complex was determined (lane 5), but rs1927914A-protein complex wasn't (lane 2). This indicated the capability of rs1927914G allele, not rs1927914A, to bind nuclear protein. This complex also can be inhibited by excess unlabeled oligonucleotide probe (lane 6).

## DISCUSSION

Because the symptoms of esophageal cancer are not obvious in the early stage, most of the patients are diagnosed in the middle and late stages and often accompanied by malnutrition. A multicenter study, which investigated the potential epidemiological and clinical risk factors affecting the survival of esophageal cancer patients in China, demonstrated that the overall 5-year survival rate is around 39% (He et al., 2020). Multiple large clinical studies
have shown that concurrent chemoradiotherapy (CCRT) can significantly improve the local control rate and the overall survival rate of esophageal cancer (*Kang et al., 2018*; *Takeda et al., 2018*). Therefore, CCRT is still the standard therapy for patients with locally advanced esophageal cancer who cannot receive or refuse surgical treatment. However, CCRT is not tolerated in patients with advanced age, severe cardiopulmonary complications or malnutrition. In the past decade, targeted therapy has brought cancer treatment into the era of precision therapy with its low toxic side effects and high therapeutic efficiency. The discovery of EGFR, ALK and other driving genes in lung cancer provides an example for targeted therapy of malignant tumors. Therefore, it is still necessary to look for potential molecular targets to guide the clinical treatment of esophageal cancer.

TLRs are important components of inflammatory response by effecting on innate immune response. So far, 10 members (TLR1-TLR10) have been identified in TLR family which involved in multiple biological processes, such as inflammatory response, immune response, apoptosis and angiogenesis and further contributed to the development of various cancers (*Belmont et al., 2014*; *Dajon, Iribarren & Cremer, 2017*; *Garcia et al., 2016*; *Paone et al., 2010*; *Vijay, 2018*). *TLR4* locates in chromosome 9q32-33. TLR4 mRNA can be polyadenylated at 3′ UTR to produce 5432nt and 12853nt transcripts that both encode the same 839aa protein. Kutikhin et al. found that the high expression of TLR4 in cancer tissues can promote the metastasis and invasion of tumor cells, and it is not suppressed by the immune system (*Davoodi, Hashemi & Seow, 2013*; *Kutikhin et al., 2014*). The overexpression of TLR4 in ESCC tissues was also associated with the poor prognosis (*Li et al., 2018*; *Sato et al., 2020*).

So far, several studies have found that TLR4 polymorphisms influence cancer susceptibility, such as gastric cancer, myeloma and hepatocellular carcinoma (*Bagratuni et al., 2016*; *He et al., 2018*; *Huang et al., 2017*). In Chinese population, Huang et al. found that there is a significantly decreased risk of gastric cancer in individuals carrying of the allele C for the rs10116253 and allele T for the rs1927911 in TLR4 (*Huang et al., 2014*). Similar results were found in hepatocellular carcinoma (*Minmin et al., 2011*). *Song et al. (2009)* found that both TLR4 rs1927911 and rs11536858 polymorphism increased the susceptibility of prostate cancer in Korean Men.

In this study, the online databases of TRANSFAC and SNPinfo Web Server were used to predict the SNPs that may affect the expression of TLR4. The prediction results showed that rs1927914 in the promoter *TLR4* affected the binding capability of the organic cation transporter 1 (Oct-1) which is a member of the POU homeodomain family of transcription factors (*Verrijzer & Van der Vliet, 1993*). The main feature of this family is its highly conserved original POU domain composed of 150 amino acids, which has a high affinity for the octamer binding sequence 5′-ATGCAAAT-3′ (*Verrijzer et al., 1992*). Studies have showed that Oct-1 was abnormally expressed in a variety of cancers and the overexpression of Oct-1 was associated with the poor prognosis in well-differentiated gastric adenocarcinoma patients (*Jeong et al., 2014*; *Rhodes et al., 2007*). In this study, our results demonstrated that TLR4 rs1927914 A>G genetic polymorphism contributed to a reduced risk of esophageal cancer. This finding was further supported by luciferase reporter assay which showed that TLR4 rs1927914 G-containing constructure displayed

higher luciferase activity than rs1927914A-containing constructe. We also found that the oligonucleotide probe with TLR4 rs1927914G could bind with the nuclear extract from esophageal cancer cells using EMSA; however, that with rs1927914A allele couldn't. There were several studies reported the association of rs1927914 polymorphism with the risk of other cancer types. For example, Shi and Minmin et al. reported that TLR4 rs1927914 genetic variations are correlated with the hepatocellular carcinoma susceptibility (*Minmin et al., 2011*; *Shi et al., 2017*). However, researchers didn't find the correlation between TLR4 rs1927914 and the risk of lung or gastric cancer (*Huang et al., 2010*; *Wu et al., 2020*). Therefore, it is suggested that TLR4 rs1927914 may be associated with the occurrence of certain cancer type. For rs11536891 polymorphisms, we predicted that it affected the binding capability of hsa-miR-519a/hsa-miR-519b-3p; however, our study didn't show this SNP on the risk of esophageal cancer. At present, there are few studies on the correlation between TLR4 rs11536891 polymorphism and cancer susceptibility. Researchers didn't find that TLR4 rs11536891 was associated with the risk of prostate cancer and lung cancer (*Song et al., 2009*; *Wu et al., 2020*). Tsilidis et al. reported that this SNP was contributed to the colorectal cancer risk (*Tsilidis et al., 2009*). These findings suggested that TLR4 might promote esophageal cancer cell proliferation through different pathways.

In addition to genetic factors, epidemiological evidence also proved that cigarette smoking strongly elevated the susceptibility to ESCC (*Abnet, Arnold & Wei, 2018*; *Chen et al., 2010*; *Dong & Thrift, 2017*). Thus, we performed stratification analysis by smoking status and found TLR4 rs1927914 GG genotype carriers had decreased risk of ESCC among non-smokers, but not among smokers. Meanwhile, we found that GG genotype is a protective factor for older subjects. These results suggest that the risk of ESCC is mainly caused by the combination of environmental and genetic factors.

## CONCLUSION

In summary, we found that rs1927914 A>G polymorphism in the promoter of TLR4 could affect the transcriptional activity of TLR4 and contributed to the susceptibility to ESCC. These data further supported the hypothesis that naturally occurring variants in innate immune genes conferred individual's susceptibility to esophageal cancer. The TLR4 polymorphism might serve as a biomarker for evaluation of esophageal cancer risk.

## ACKNOWLEDGEMENTS

The authors thank all patients and control subjects for their participation.

### Funding

This work was supported by the Key Project of Natural Science Foundation of Hebei province of China (H2017209233 to Xuemei Zhang), the Leader talent cultivation plan of innovation team in Hebei province (LJRC001 to Xuemei Zhang) and the National Natural Science Foundation of China (81101483 to Xuemei Zhang). There was no additional

external funding received for this study. The funders had no role in study design, data collection and analysis, decision to publish, or preparation of the manuscript.

## Grant Disclosures

The following grant information was disclosed by the authors:
Key Project of Natural Science Foundation of Hebei province of China: H2017209233.
Leader talent cultivation plan of innovation team in Hebei province: LJRC001.
National Natural Science Foundation of China: 81101483.

## Competing Interests

The authors declare there are no competing interests.

## Author Contributions

- Jiaying Li performed the experiments, analyzed the data, prepared figures and/or tables, authored or reviewed drafts of the paper, and approved the final draft.
- Hongjiao Wu, Hui Gao, Yuning Xie and Zhi Zhang analyzed the data, prepared figures and/or tables, and approved the final draft.
- Ruihuan Kou performed the experiments, analyzed the data, prepared figures and/or tables, and approved the final draft.
- Xuemei Zhang conceived and designed the experiments, authored or reviewed drafts of the paper, and approved the final draft.

## Human Ethics

The following information was supplied relating to ethical approvals (i.e., approving body and any reference numbers):

This research was approved by Institutional Review Board of North China University of Science and Technology had approved the (12-002).

## Data Availability

Raw data are available in the Supplemental Files.

## Supplemental Information

Supplemental information for this article can be found online at http://dx.doi.org/10.7717/peerj.10754#supplemental-information.

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
