# Peer review of "TLR4 promoter rs1927914 variant contributes to the susceptibility of esophageal squamous cell carcinoma in the Chinese population"

_PeerJ, doi:10.7717/peerj.10754_

## Round 0.1 · original submission · Major Revisions

Please address all the critiques raised by the reviewers and revise manuscript accordingly.

Reviewer 1 ·

Basic reporting

This manuscript meets the basic criteria of reporting.

Experimental design

The experimental setup is sound and reasonable. The methods are described in details (with a few exceptions; see specific comments) and all the relevant details are provided for data transparency and reproducibility.

Validity of the findings

The findings are novel and would benefit the field.

Additional comments

The manuscript submitted by Li and colleagues, presents analytical study on the association between genetic variations (SNPs) in the Toll-like receptor 4 (TLR4) and the risk of esophageal squamous cell carcinoma (ESCC) by employing case-control research strategy. Authors provide evidence supporting that TLR4 rs1927914 variant contributes to the ESCC risk possibly by alterations in regulation of the promoter activity. Although the manuscript describes some interesting findings; there are some limitations in its current version. Here are my specific comments that may help authors to improve the quality of the manuscript.

1) Authors claim that rs1927914 G allele was associated with elevated transcriptional activity. But the experimental biochemical evidence to support this claim is missing. Can authors provide any evidence on DNA binding activity of the nuclear protein factors to these alleles suggesting differential binding pattern?
2) A schematic showing TLR4 locus with SNPs investigated in this study would help. Also, authors may use this to explain the cloning strategy used to generate luciferase reporters.
3) Line 161-162: Authors write – “Table 2 showed the association of TLR4 rs1927914, rs11536891 and rs7873784 genotypes with the susceptibility to esophageal cancer”. But the Table 2 does not have any data for rs7873784. This need correction.
4) Lines 168-170: Authors write – “After genotyping TLR4 rs7873784 (G>C) and rs11536891 (T>C)… conforms to complete genetic linkage.” These results are not citing the relevant table/s.
5) Can the authors provide an evidence [or comment] on the specificity of Allele-specific TaqMAN genotyping probes?
6) Authors used TRANSFAC program for transcription factor binding capability. The results of this analysis are not shown or discussed. Also, authors need to cite the relevant references (is it PMID: 8594589?).
7) Same applies to microRNA binding ability prediction using ‘Web Server’(lines 102-103). Which ‘web server’ was used? Please cite relevant references. Also, the results are neither shown nor discussed.
8) Authors may need to include more literature on the SNPs and their association with disease. Also, the section described in lines 78-81 needs relevant citations.
9) In the conclusion section (which is very short), authors may add a note on the future prospect of this study, how does this study help in advancement in the current field.
10) Grammar check would help e.g. Line 216: “…be polyadenylation at 3’UTR..” should read “…be polyadenylated at 3’UTR…”
11) Table 4 is actually a figure. Please correct this. The legend needs to be added for this one. Explain how was the normalization done? Also, please correct the Y axis label. “Realitive” should read “Relative”.

Reviewer 2 ·

Basic reporting

In the study by Li et al., evaluated the association of TLR4 polymorphisms with the genetic susceptibility to esophageal squamous cell carcinoma. The authors identify that the TLR4 promoter variant ‘ rs1927914’ that confer susceptibility to esophageal squamous cell carcinoma in Chinese population.

The objectives of the study are well defined and the authors provide reasonable findings to support the hypothesis.
The literature is well referenced and relevant to the study
The overall manuscript is well written, data presented is clear, conclusions are well drawn, and provides a new research finding on TLR4 polymorphism and susceptibility to ESCC.

Experimental design

In general experimental approaches are well designed with relevant controls, and appropriate statistics is included.

Some minor changes/modifications suggested as follows:
1) Line #99 (Materials-methods section), authors mention of the criteria sought to select SNPs in TLR4 promoter. This description needs to be more clarified. Authors should provide rationale for the analysis of transcription factor binding on TLR4 promoter by TRANSFAC. Which transcription factor(s) binding to TLR promoter were analyzed?
(2) Does the SNP in the TLR4 promoter, affect the binding of any known transcription factor?
(3) Line #220, in discussion section, authors mention “Therefore, reasonable inhibition of TLR4 gene expression is conducive to disease recovery…”. This sentence needs to be modified, as inhibition of a single gene may not be sufficient for the disease recovery.

Validity of the findings

The data presented and findings from the study are appropriately given and conclusions are well drawn.

Additional comments

No comment.

---

## Round 0.2 · accepted · Accept

Both reviewers are satisfied by the revision and I am happy to accept your manuscript.

Reviewer 1 ·

Basic reporting

Manuscript meets basic criteria of publication.

Experimental design

Valid experimental design and detailed description of procedures.

Validity of the findings

Manuscript provides insightful findings pertaining to esophageal squamous cell carcinoma biology.

Additional comments

Authors have addressed my comments successfully. The revised manuscript is scientifically improved and is now suitable for publication. I have only one minor suggestion. It would be helpful (for general understanding) to add an arrow pointing at the DNA-protein complex in Figure 1C (since there is no clear intact band).

Reviewer 2 ·

Basic reporting

This study aims to analyze the association of potential functional genetic polymorphisms in TLR4 with the risk of esophageal cancer.

Experimental design

Well defined, and organized.

Validity of the findings

Authors addressed the required concerns in the revised version of the manuscript.